# Flap Extension Technique Using Intrasocket Granulation Tissue in Peri-Implant Osseous Defect: Case Series

**DOI:** 10.3390/medicina58111555

**Published:** 2022-10-29

**Authors:** Won-Bae Park, Jung-Min Ko, Ji-Young Han, Philip Kang

**Affiliations:** 1Department of Periodontology, School of Dentistry, Kyung Hee University, Private Practice in Periodontics and Implant Dentistry, Seoul 02447, Korea; 2Division of Periodontics, Section of Oral, Diagnostic and Rehabilitation Sciences, Columbia University College of Dental Medicine, New York, NY 10032, USA; 3Department of Periodontology, Division of Dentistry, College of Medicine, Hanyang University, 222-1 Wangsimni-ro, Seongdong-gu, Seoul 04763, Korea

**Keywords:** dental implants, socket graft, surgical flap, granulation tissue, bone regeneration, surgical closure technique

## Abstract

A compromised extraction socket is characterized by severe bone resorption around neighboring teeth and is often occupied with thick intrasocket granulation tissue (IGT). Guided bone regeneration (GBR) is a procedure that can preserve the bone volume around extraction sockets, and it can also be combined with immediate implant placement. However, an early exposure of GBR sites is a possible complication because it increases the risk of infection and can inhibit successful bone regeneration. The purpose of these case series is to introduce a novel, surgical procedure that can prevent the exposure of GBR sites by using IGT for flap extension during immediate implant placement in compromised extraction sockets. The technique was successfully performed in six patients. For successful flap closure, the inner portion of the IGT was dissected so that the flap was properly extended with the base of IGT attached to the flap for blood supply. Periosteal releasing incisions were not performed. The IGT was first sutured to the palatal flap with resorbable sutures, and then the overlying flap was closed with additional sutures. There was no post-operative exposure of the surgical GBR site in any of the patients, and the location of the mucogingival junction remained unchanged. All grafted sites also achieved sufficient bone regeneration. Within the limitations, this case series demonstrates the potential use of IGT, a concept which was previously obsolete.

## 1. Introduction

Guided bone regeneration (GBR) is a predictable concept that can be applied in the treatment of various peri-implant osseous defects including in extraction sockets during immediate implant placement. GBR uses bone grafts for space maintenance and barrier membranes for epithelial cell exclusion [1,2]. Various factors can affect the outcome of the GBR procedure including the choice of bone graft and barrier membrane, the size and shape of the defect, the thickness of the flap and the achievement of primary closure, and the operator’s clinical experience level [3]. In particular, the sophistication of flap management by the operator is of great importance to the success of GBR procedures, especially in cases of immediate implant placement as achieving tension-free primary closure can be challenging due to limited soft tissue volume.

Immediate implant placement is often performed due to its many advantages over delayed implant placement, especially in the maxillary anterior region. Immediate placement minimizes resorption of the buccal bone plate and enables successful bone augmentation [3,4]. On the other hand, delayed placement has its advantage of allowing better flap closure; however, several additional procedures are sometimes required to restore the esthetics due to soft tissue or hard tissue deficiencies after resorption of the thin buccal bone plate. Therefore, although technique-sensitive, immediate placement can be considered preferentially for implant placement after extraction in the esthetic zone.

Primary closure in GBR of extraction sockets can be achieved with several techniques like using periosteal releasing incisions [5], a combination of periosteal releasing incisions and vertical incisions, or a soft tissue graft [6]. However, these techniques may cause unfavorable results post-operatively including the shallowing of the vestibular depth and the decrease in the width of keratinized mucosa. Technical modifications in the GBR procedure have been proposed to compensate for these shortcomings including using granulation tissue of the extraction socket for flap extension. Mardinger et al. [7,8] and Hur et al. [9] reported that intrasocket granulation tissue (IGT) in a compromised extraction socket could be used to make primary closure possible during ridge augmentation at time of implant placement or ridge preservation before implant placement without changing the vestibular depth.

It is also a common belief that it is not safe to place the implant immediately into a periodontally compromised socket because of the lack of sufficient bone. Mardinger et al. [8] reported that implant placement was performed 6 months after ridge preservation of a periodontally compromised socket. Additionally, An et al. [10] reported that a compromised socket of a premolar or molar achieved sufficient vertical and horizontal bone gain one year after extraction. These suggest that a compromised socket takes a long time to heal and achieve suitable dimensions for implant placement if only ridge preservation is completed. 

From the knowledge of the authors, the surgical technique of utilizing intrasocket granulation tissue described in this case report has not been widely recognized among clinicians. However, to increase the success of GBR cases around compromised extraction sockets, the surgical site must remain unexposed and primary closure must be maintained until the time of uncovering. This case report introduces the flap extension method using the IGT of a compromised extraction socket for successful primary closure and evaluates its clinical and radiological results in six patients.

## 2. Detailed Case Description

Six patients underwent extraction, immediate implant placement, and guided bone regeneration (GBR) using intrasocket granulation tissue (IGT) for flap extension and primary closure. Patient information including age, sex, smoking status, site of compromised socket, implant information, and follow up period is shown in Table 1. 

The surgical method used in each patient is as follows:
A tooth or implant with severe bone loss was removed using extraction forceps under local anesthesia with 2% lidocaine containing 1:100,000 epinephrine;A midcrestal incision and two buccal vertical incisions were made. Before raising the flap, special care was taken to ensure a clean dissection of the IGT from the underlying flap without any damage and the IGT is still well-attached to the flap. The IGT was then further separated from the bony housing by using a curette with various curvatures and a periosteal elevator.The IGT attached to the buccal flap was dissected to an appropriate thickness using a #15 blade so that the base of the IGT was still attached to the existing flap, and the flap was eventually extended. Periosteal releasing incisions were not performed at the flap base.The extraction socket was thoroughly debrided using a Molt curette and a titanium brush.A surgical guided stent was used so that the implant (Implantium, Dentium, Suwon, Korea) was placed 2.0 mm subcrestal to the level of the adjacent bone in the extraction socket.A synthetic osteoconductive bone graft substitute composed of hydroxyapatite (HA) and beta-Tricalcium phosphate (β-TCP) (Osteon III, Genoss, Suwon, Korea) and a resorbable collagen membrane (Genoss, Suwon, Korea) were placed to cover the implant and the peri-implant osseous defect.After covering the bone graft with the collagen membrane, the extended IGT was sutured with the palatal flap with 4-0 Catgut. Next, the buccal flap was closed using 4-0 nylon or black silk.Antibiotics (Cefradine 500 mg, Yuhan Pharmaceutical Co., LTD. Seoul, Korea) and anti-inflammatory drugs (Etodol^®^ 200 mg, Yuhan Pharmaceutical Co., LTD. Seoul, Korea) were prescribed for 10 days. The patient was recommended to use 0.12% chlorhexidine solution (Hexamedine, Bukwang Pharmaceutical, Seoul, Korea) twice a day for two weeks. Sutures were removed after 10 days.Uncovering procedures were performed 4–6 months after initial surgery. Under local anesthesia, the buccal flap was reflected, the regenerated tissue above the implant cover screw was removed, and the healing abutment was connected to the implant. The buccal flap was closed using 4-0 Catgut or black silk. Antibiotics and anti-inflammatory drugs were prescribed for 5 days. The prosthesis was installed 2 months after uncovering.

## 3. Case 1

Patient #1 was a 52-year-old non-smoker male with no systemic conditions affecting the operation. The patient visited the clinic due to severe mobility of the maxillary central incisor. This case is depicted in Figure 1. In the preoperative panoramic radiograph (Figure 1a), severe bone resorption was observed around the root of #21 with periapical radiolucency. In the CBCT cross-sectional view (Figure 1b), IGT was observed around the root of #21. #21 had probing depths of more than 6 mm at all surfaces and grade II tooth mobility. Extraction, immediate implant placement and GBR were performed according to the described surgical protocol (Figure 1d–h). After extraction of #21, buccal and palatal flaps were reflected and thorough defect debridement was performed. The IGT attached to the buccal flap was dissected so that the base was attached to the existing flap using a #15 blade, and the flap was extended (Figure 1d). The implant placed at #21 was a 3.8 mm× 12 mm Implantium (Dentium, Suwon, Korea). A large peri-implant osseous defect occurred after implant placement (Figure 1e). Bone graft substitute (Osteon III, Genoss, Suwon, Korea) was placed to cover the implant (Figure 1f). A resorbable collagen membrane (Genoss, Suwon, Korea) was placed over the graft and the extended IGT was sutured with the palatal flap with resorbable sutures (Figure 1g). Primary closure was achieved with the extended IGT (Figure 1h). A removable provisional restoration was delivered 2 weeks after surgery. The GBR site healed well without any exposure (Figure 1i). An uncovering procedure was performed 6 months after surgery with two submarginal vertical incisions and buccal flap reflection. The observed regenerated tissue was very dense. The final prosthesis was delivered 2 months after the uncovering procedure (Figure 1j). In the panoramic radiograph and CBCT scan taken 12 months after the prosthesis was delivered (Figure 1k–l), an adequate amount of new bone was observed on the labial surface of the implant.

## 4. Case 2

Patient #2 was a 53-year-old non-smoker female with no systemic conditions affecting the operation. The patient visited a private clinic seeking to replace her recently extracted #23 implant. This case is depicted in Figure 2. The preoperative panoramic radiograph and CBCT scan showed severe bone resorption and IGT around the extracted #23 implant socket. Implant placement and GBR were performed according to the surgical protocol as described above. The GBR site healed well without exposure. An uncovering procedure was performed 6 months after the implant surgery. After tissue punching, the healing abutment was inserted and the final prosthesis was installed after 6 weeks. A panoramic radiograph and CBCT scan 29 months after prosthesis delivery showed that the marginal bone level was well maintained on implant #23.

## 5. Case 3

Patient #3 was a 72-year-old smoker male taking antihypertensive drugs and antithrombotic drugs. This case is depicted in Figure 3. The preoperative panoramic radiograph and CBCT scan showed severe bone resorption around the root of #13 and IGT was observed. Clinical exam showed a deep probing depth and severe tooth mobility on #13. A 3.8 mm × 12 mm Implantium implant (Dentium, Suwon, Korea) was placed in the extraction socket of #13, and surgery was performed according to the surgical protocol. The GBR site healed well without exposure. An uncovering procedure was performed 6 months after surgery and the observed regenerated tissue was very dense. The final prosthesis was delivered after 6 weeks. In the panoramic radiograph and CBCT scan 35 months after prosthesis delivery, the peri-implant osseous defect was well-regenerated.

## 6. Case 4

Patient #4 was a 76-year-old non-smoker female patient with no systemic conditions affecting the operation except for her rhinitis. The preoperative panoramic radiograph and CBCT showed severe bone resorption and IGT around existing #13 implant. After extraction of failing implant #13, a 3.8 mm × 12 mm Implantium implant (Dentium, Suwon, Korea) was placed (Figure 4). Bone graft and collagen membrane placement and flap closure with IGT extension were performed according to surgical protocol. The GBR site healed well without exposure. The uncovering procedure was performed 6 months after surgery. The regenerated tissue was very hard. 6 weeks after the uncovering procedure, the final prosthesis was delivered. It was confirmed that all peri-implant osseous defects were regenerated in the panoramic radiograph and CBCT scan 28 months after the prosthesis was delivered.

## 7. Case 5

Patient #5 was a 54-year-old smoker male with had severe bone resorption around #36 (Figure 5). Extraction, implant placement and simultaneous GBR were performed according to the surgical protocol. The GBR site healed without exposure and the location of the mucogingival junction was also unchanged. After 5 months, an uncovering procedure was confirmed that new bone was well formed at the GBR site. In the panoramic radiograph 19 months after prosthesis placement, there was no change in crestal bone level around the implant.

## 8. Case 6

Patient #6 was a 72-year-old non-smoker female who visited the clinic for implant placement in the left posterior mandible (Figure 6). The preoperative panoramic radiograph showed severe bone resorption around teeth #35 and #36. Implants were placed in compromised sockets after extractions of #35 and #36, and peri-implant osseous defects were treated according to the surgical protocol. The GBR sites were not exposed and the location of the mucogingival junction was not changed. After 4 months, an uncovering procedure was performed and it was observed that the bone defects were filled with dense bone. 26 months after the delivery of the final prosthesis, there was no change in crestal bone level around implant #35.

## 9. Discussion

In the case of a tooth or implant with severe infection, infectious granulation tissue is formed. This IGT was referred to as “intrasocket reactive soft tissue” in previous studies [7,8] and this granulation tissue was removed during socket preservation or socket augmentation because it contains many inflammatory cells and long junctional epithelium [8]. However, Mardinger et al. [8] and Hur et al. [9] suggested that IGT can play a positive role in GBR procedures. This case series demonstrated that primary closure can be achieved by extending the flap through dissection of IGT after immediate implant placement in a compromised extraction socket. The wound edges of all cases were well closed without membrane exposure even though periosteal releasing incisions were not performed. In addition, this procedure resulted in little shift in the location of the mucogingival junction, and there were no severe post-operative complications. Sufficient bone regeneration at the surgical sites was also well observed.

Compared to extraction sockets of uncompromised teeth, compromised extraction sockets have larger bone defects and more granulation and inflammatory tissues. Kim et al. [6] stated that it is important to completely remove the infection source during ridge preservation in a compromised extraction socket. Ridge augmentation after sufficient soft tissue healing or delayed implant placement after ridge preservation are common techniques that can be applied in cases of compromised sockets [6,8,9,10]. However, natural, non-intervened healing after extraction of periodontally compromised teeth leads to severe loss of both soft and hard tissues. According to a systematic review by Tan et al. [11], after tooth extraction, there is rapid loss of alveolar bone in the first 3–6 months and gradual reduction thereafter. Therefore, delayed implant placement performed after healing may require a large amount of GBR. Esthetic problems may also occur after surgery with soft and hard tissue deficiencies especially in the esthetic area. In this respect, the treatment combining immediate implant placement and simultaneous GBR can have many advantages.

Maintaining a tension-free flap for primary closure is a critical factor for the success of a GBR procedure. Fugazzotto [12] suggested that membrane exposure occurring within 6 months after GBR therapy is considered a failure. Membrane exposure can increase the risk of infection and damage to bone formation [13]. However, there are studies that show that bone regeneration is not affected even with an open wound, and there are techniques such as the open membrane technique to intentionally expose the GBR technique [14,15,16]. The low porosity of the d-PTFE (dense polytetrafluroethylene) membrane is resistant to bacterial infiltration, reducing exposure problems [17]. On the other hand, the resorbable collagen membrane is continuously absorbed and incorporated into the host tissue. Therefore, it is expected that there will be little or no adverse effects from membrane exposure of a collagen membrane [18,19]. However, according to the authors’ experience and several reports, it is true that the exposure of the wound edge can have a high risk of infection and insufficient bone regeneration [13,20]. Sbricoli et al. [21] stated that in clinical practice, the healing process after application of the collagen membrane is uneventful, but the observation of membrane exposure is not uncommon; in addition, they described that when the membrane was exposed to the oral environment, it had a high risk of bacterial colonization, resulting in faster degradation and ultimately resulting in severely reduced regeneration. Therefore, several surgical techniques have been introduced to enhance the closure of the flaps. Commonly used methods include periosteal releasing incisions, horizontal mattress sutures, double flap incision, and addition of a subepithelial connective tissue graft [22,23,24,25]. These techniques can lead to postoperative complications such as bleeding, swelling, and hematomas and may require additional treatment related to this [26,27].

In the reported cases, although periosteal releasing incisions were not performed to help with primary wound closure, early wound exposures did not occur because of the utilization of the IGT that can provide additional soft tissue support. Furthermore, there was no intentional flap advancement, so there was no change in vestibular depth and no loss of keratinized gingiva. This technique is suitable for a compromised socket with a IGT that is thick (more than 2 mm) and wide. If the IGT is thin, it is more likely to be damaged during incision and dissection for flap extension. Therefore, this procedure is suitable to be performed in a compromised extraction socket with more severe bone defects. However, one must be cautious about the interpretation of the described technique. The definition of granulation tissue is histological and, from a clinical point of view, one could only assume to cut exactly between the granulation tissue and the flap, thus, from time to time, a unintentional semi-split-thickness flap may occur. The disadvantage of this case report is its limited patient pool. A study on the clinical effect and validity of the procedure will be needed in the future.

## 10. Conclusions

Within the limitations of this case series, the IGT of compromised extraction sockets is potentially useful for primary closure of the wound when performing GBR surgery concurrently with immediate implant placement.

## Figures and Tables

**Figure 1 medicina-58-01555-f001:**
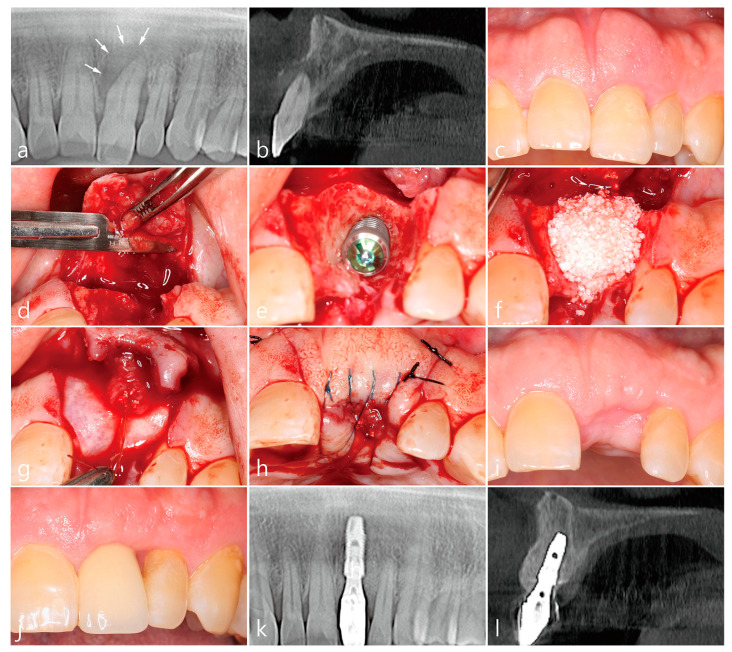
Clinical and radiological findings of case #1. (**a**) Pre-operative panoramic radiograph showing severe bone resorption and periapical radiolucency around the root of #21 (white arrow). (**b**) Cross-sectional image of CBCT #21 showing ICT around #21. (**c**) Pre-operative photograph #21. (**d**) Flap extension by dissection of the ICT attached to the buccal flap with the base attached to the existing flap using a #15 blade. (**e**) Implant #21 placement with peri-implant osseous defect. (**f**) Bone graft placement over implant. (**g**) Resorbable collagen membrane placement and extended IGT sutured to the palatal flap with resorbable sutures. (**h**) Primary closure with extended IGT. (**i**) Post-operative photograph showing no flap exposure. (**j**) Final prosthesis #21. (**k**) Panoramic radiograph taken 12 months after prosthesis delivery. (**l**) Cross-sectional image of CBCT taken 12 months after prosthesis delivery with sufficient bone regenerated around the implant.

**Figure 2 medicina-58-01555-f002:**
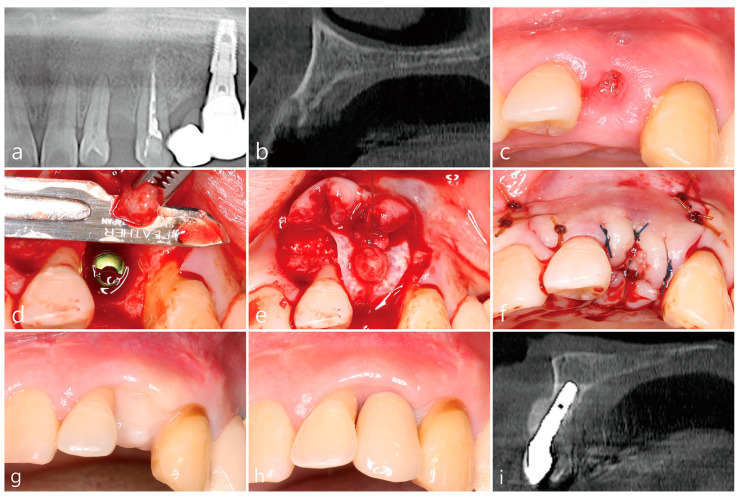
Clinical and radiological findings of case #2. (**a**) Pre-operative panoramic radiograph showing compromised extraction socket #23. (**b**) Cross-sectional image of CBCT #23 extraction socket showing loss of labial bone plate with ICT. (**c**) Pre-operative photograph showing extraction socket #23 (extracted prior to surgery). (**d**) Flap extension by dissection of the ICT attached to the buccal flap with the base attached to the existing flap after implant placement. (**e**) Bone graft and resorbable collagen membrane placement and extended IGT sutured to the palatal flap with resorbable sutures with 4-0 Catgut. (**f**) Flap closure with extended IGT. (**g**) Post-operative photograph showing healing with no flap exposure and sufficient amount of keratinized gingiva. (**h**) Final prosthesis #23. (**i**) Cross-sectional image of CBCT taken 29 months after prosthesis delivery with sufficient bone regenerated around the implant.

**Figure 3 medicina-58-01555-f003:**
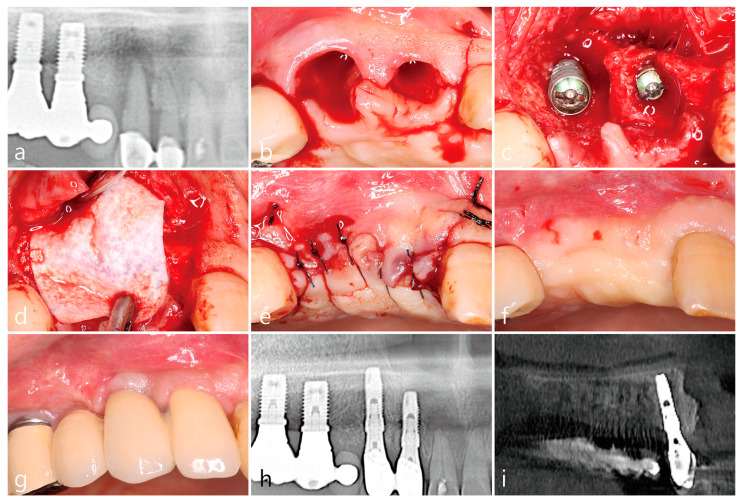
Clinical and radiological findings of case #3. (**a**) Pre-operative panoramic radiograph showing severe bone resorption and IGT around #13. (**b**) Mid-crestal and two vertical incisions after tooth extraction. (**c**) Implant #12 and #13 placement with peri-implant osseous defects. (**d**) Bone graft and resorbable collagen membrane placement over implants. (**e**) Flap closure with extended IGT. (**f**) Post-operative photograph showing healing with no flap exposure. (**g**) Final prosthesis #12 and #13. (**h**) Panoramic radiograph taken 35 months after prosthesis delivery. (**i**) Cross-sectional image of CBCT taken 35 months after prosthesis delivery with sufficient bone regenerated around the implant.

**Figure 4 medicina-58-01555-f004:**
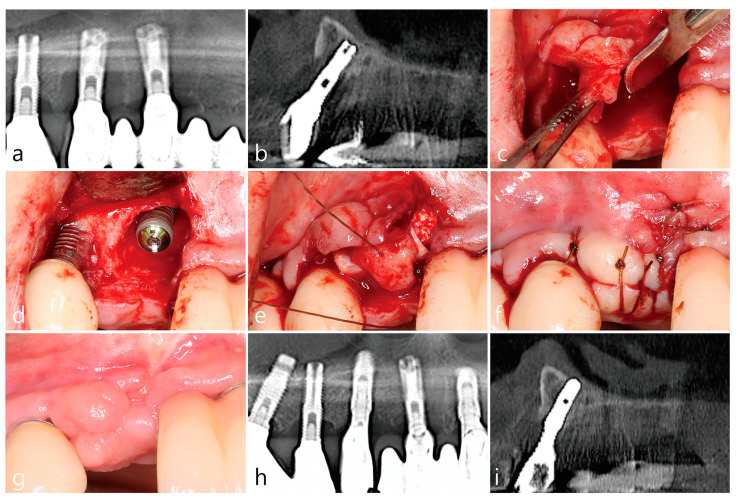
Clinical and radiological findings of case #4. (**a**) Pre-operative panoramic radiograph showing severe bone resorption and periapical radiolucency around the existing implant #13. (**b**) Cross-sectional image of CBCT #13 showing ICT. (**c**) Flap extension by dissection of the ICT attached to the buccal flap with the base attached to the existing flap using a #15 blade. (**d**) Implant #13 placement with peri-implant osseous defect. (**e**) Bone graft and resorbable collagen membrane placement over implant. Extended IGT sutured to palatal flap. (**f**) Flap closure with extended IGT. (**g**) Post-operative photograph showing healing with no flap exposure. (**h**) Panoramic radiograph taken 28 months after prosthesis delivery. (**i**) Cross-sectional image of CBCT taken 28 months after prosthesis delivery with sufficient bone regenerated around the implant.

**Figure 5 medicina-58-01555-f005:**
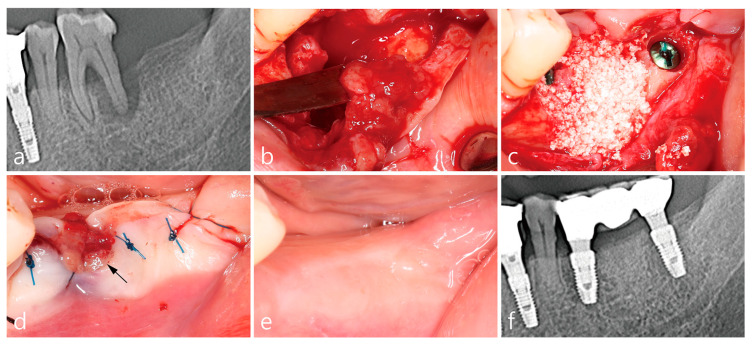
Clinical and radiological findings of case #5. (**a**) Pre-operative panoramic radiograph showing severe bone resorption around #36. (**b**) Flap extension by dissection of the ICT attached to the buccal flap with the base attached to the existing flap. (**c**) Bone graft placement over #36 implant. (**d**) Flap closure with extended IGT. (**e**) Post-operative photograph showing healing with no flap exposure. (**f**) Panoramic radiograph taken 19 months after prosthesis delivery.

**Figure 6 medicina-58-01555-f006:**
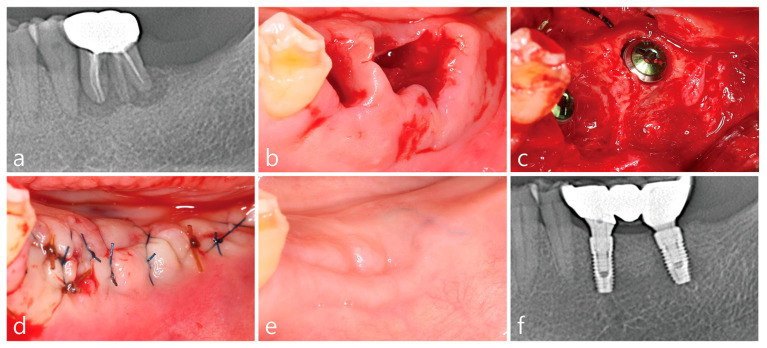
Clinical and radiological findings of case #6. (**a**) Pre-operative panoramic radiograph showing severe bone resorption around #35 and 36. (**b**) Compromised sockets #35, 36. (**c**) #35, 36 implant placement with peri-implant osseous defect. Flap extension by dissection of the ICT attached to the buccal flap with the base attached to the existing flap. Bone graft placement over #36 implant. (**d**) Flap closure with extended IGT. (**e**) Post-operative photograph showing healing with no flap exposure. (**f**) Panoramic radiograph taken 26 months after prosthesis delivery.

**Table 1 medicina-58-01555-t001:** Patient information.

Case	Age/Sex	Smoking	Compromised Socket Site	Implant Size	Follow-Up Period (Months)
1	52/M	No	#21	3.8 × 12	12
2	53/F	No	#23	3.8 × 10	29
3	72/M	Yes	#13	3.8 × 12	35
4	76/F	No	#13	3.8 × 12	28
5	54/M	Yes	#36	4.3 × 10	19
6	72/F	No	#35/#36	4.3 × 10	26

## Data Availability

All data is contained within the article.

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
