# Peer review of "Flap Extension Technique Using Intrasocket Granulation Tissue in Peri-Implant Osseous Defect: Case Series"

_medicina, 2022, doi:10.3390/medicina58111555_

Round 1

Reviewer 1 Report

Overall, the paper is well written. I present minor modifications and major issues:

Minor modifications:

Pag. 2 - line 66: please improve the English. Difficult to understand. "... reported that implant placement was performed 6 months after ridge preservation..." ; I believe the author want to say: Difficult to understand. "... reported that implant placement should be performed 6 months after ridge preservation...". Please correct.

Pag.2 - Line 68: "1 year" ; please modify to "one year". Please correct.

Pag.3 - Line 122. The author wrote "ICT". I believe it should be "IGT". Please correct.

Pag. 6 - line 193 and 194. It seems that the author uses ADA tooth numbering system and FDI tooth numbering system at the same time. Canine is #6 and #13... please correct.

MAJOR ISSUES:

1 - The introduction lacks a paragraph about the biology of the IGT. There's only one sentence about IGT in pag.2 line 59-62/63. The author should write a paragraph about IGT (in terms of biology and healing) because this is a controversial issue. Also, this should be properly addressed in the discussion. Why - for decades - we have learned to remove this granulation tissue to promote healing, and now you are proposing to use it in complex cases....

2 - In none of the cases it is presented the periodontal pocket depth. In a paper on this theme, the periodontal pocket depth should be evaluated and presented mainly after the initial evaluation of the case, but also in the follow-up appointment.

3 - Pag.7, case n.º 4 - in fig 4 f) it seems that the primary closure was completely achieved with the flap. I don't see any exposure of the IGT as in the other cases. Can you explain? The same in case 6. Can you explain?

--------

For a controversial theme, all the presented cases showed a good follow-up. Congratulations on that. However, the authors should discuss more why this granulation tissue is not affecting/infecting the GBR.

Author Response

Thank you very much for your comments.  Below are the authors' point-by-point responses to address your concerns.

Reviewer 1:

 Minor modifications:

Pag. 2 - line 66: please improve the English. Difficult to understand. "... reported that implant placement was performed 6 months after ridge preservation..." ; I believe the author want to say: Difficult to understand. "... reported that implant placement should be performed 6 months after ridge preservation...". Please correct.

Revised.

Pag.2 - Line 68: "1 year" ; please modify to "one year". Please correct.

 Corrected.

Pag.3 - Line 122. The author wrote "ICT". I believe it should be "IGT". Please correct.

Corrected.

Pag. 6 - line 193 and 194. It seems that the author uses ADA tooth numbering system and FDI tooth numbering system at the same time. Canine is #6 and #13... please correct.

Corrected.

MAJOR ISSUES:

1 - The introduction lacks a paragraph about the biology of the IGT. There's only one sentence about IGT in pag.2 line 59-62/63. The author should write a paragraph about IGT (in terms of biology and healing) because this is a controversial issue. Also, this should be properly addressed in the discussion. Why - for decades - we have learned to remove this granulation tissue to promote healing, and now you are proposing to use it in complex cases....

The authors believe that the use of intrasocket granulation tissue for flap closure is not a widely accepted practice and the biological evidence is lacking; however, the technique can be useful in limited cases where the IGT is thick and without inflammation. A few sentences in the introduction (lines 71-75) and discussion (lines 292-297) have been added to address your concern.  Thank you.

2 - In none of the cases it is presented the periodontal pocket depth. In a paper on this theme, the periodontal pocket depth should be evaluated and presented mainly after the initial evaluation of the case, but also in the follow-up appointment.

Thank you for pointing this out.  As the objective of the case report was to evaluate the healing of implant surgical sites after using intrasocket granulation tissue, details on pre- and post-operative periodontal status have not been added, expect for case 1 where more detailed information has been added (lines 118-119).

3 - Pag.7, case n.º 4 - in fig 4 f) it seems that the primary closure was completely achieved with the flap. I don't see any exposure of the IGT as in the other cases. Can you explain? The same in case 6. Can you explain?

For all cases, IGT was placed under the flap and primary closures were achieved.  There isn't any exposure of IGT in any of the reported cases.

--------

For a controversial theme, all the presented cases showed a good follow-up. Congratulations on that. However, the authors should discuss more why this granulation tissue is not affecting/infecting the GBR.

Thank you for your comments.  As this topic has not been investigated fully and it still remains controversial, the authors could not provide any scientific rationale behind this idea.  However, proposed ideas and available literature have been added in lines 244-248.    

Reviewer 2 Report

Attached file

Author Response

Thank you very much for your comments.  Below are the authors' point-by-point responses to address your concerns.

Comments to Authors

Title:

- The title makes it evident that the flap extension technique using intrasocket granulation tissue was performed to prevent the exposure of the collagen membrane. However, during the article, only in the discussion is it mentioned the importance of avoiding the exposure of the collagen membrane in the surgical procedure. I suggest changing the title of the article to "Flap Extension Technique Using Intrasocket Granulation Tissue in Peri-Implant Osseous Defect. Case Series" or I would suggest inserting a paragraph in the introduction and adding this point to the objectives of the article (L72-78). (L72-78).

The title has been corrected.

Key-words:

Dental implants: Ok.
Extraction socket: No items found. (Changed to Socket Graft)

Flap closure: No items found. (Changed to Surgical Flap)
Granulation tissue: Ok
Guided bone regeneration: No items found. (Changed to Bone Regeneration)

Primary closure: No items found. (Changed to Surgical Closure Technique)

Introduction

- The authors need to clarify what is unique and special about this case series and what they add to the scientific literature.

Two sentences have been revised and added to address this issue (lines 71-75)

Page 2 (line 57)
“the width of keratinized gingiva” change to “the width of keratinized mucosa”

Corrected.

Detailed Case Description

- This reviewer strongly recommends that authors follow the CARE guidelines in reporting this clinical case series. The CARE guidelines (for CAse REports) were developed by an international group of experts to support an increase in the accuracy, transparency, and usefulness of case reports.

The submitted case report follows the CARE guidelines.

- Was informed consent obtained to perform the procedure and take the photographic images? If so, I suggest including this information.

Yes, the obtained informed consents were submitted, but they would not be included in the actual publication.

- Page 2 (lines 85 and 86)
What concentration of the local anesthetic lidocaine was used?

2% Lidocaine 1:100,00 epinephrine has been added to the text (line 85)

- Page 3 (lines 97 and 98)
Please include more details regarding the biomaterials used (bone substitute and collagen membrane)

More details have been added. (lines96-97)

- Page 3 (lines 102 and 103)
I would like the authors to justify using antibiotics in these case reports. The literature does not support the use of antibiotics for dental implant installation.
In a systematic review published in "Medicina”:

Singh Gill A, Morrissey H, Rahman A. A Systematic Review and Meta-Analysis Evaluating Antibiotic Prophylaxis in Dental Implants and Extraction Procedures. Medicina (Kaunas). 2018 Dec 1;54(6):95. doi: 10.3390/medicina54060095. PMID: 30513764; PMCID: PMC6306745.

Conclusion: “No statistically significant evidence was found to support the routine use of prophylactic antibiotics in reducing the risk of implant failure or post-operative complications under normal conditions.”

“Based on the articles analysed in this review it is recommended that clinicians carefully consider the appropriate use of antibiotics in dental implants and extraction procedures even if it is financially feasible due to risk of allergic/toxic reactions and the development of antibiotic resistance. Further monitoring of antibiotic prescribing in dentistry is required in addition to continuing education for dentists concerning the public health risks associated with antibiotic misuse.”

- Page 3 (line 110)
I think the authors used antibiotics unnecessarily. They contradict the study published in 2018 in this same scientific journal. Again, the authors use antibiotics for a simple surgical procedure (uncovering). Therefore, a justification is critical, as this conduct may lead dental professionals worldwide to prescribe antibiotics for surgical procedures like the ones described here.

Thank you very much for pointing out this controversial topic.  As much as our authors agree with your statement, there isn't any specific guideline or standard that clinicians should follow.  The decision is made mainly based on each clinician's judgment.

- Page 3 (line 119)

“#21 had a deep probing depth and severe tooth mobility.”

The authors mention that the tooth had deep probing depth and severe tooth mobility. The deep probing depth and severe tooth mobility are imprecise terms. Please include precisely the clinical probing depth on the different tooth surfaces (MB, B, DB, ML, L, DL) and the degree of tooth mobility (grade I, grade II, or grade III).

Corrected.

I recommend that the authors include in all clinical cases the postoperative follow-up periods. The authors should make it clear if there were any postoperative complications.

All pre- and post-op information is included in each case description.

Discussion

- Authors should begin the discussion by describing the main results of the case series and only then support it with the literature.

As the main results have already been discussed in detail in each case description immediately before the discussion section, the authors believed that duplication of such information section was considered redundant. 

- What is the study's new contribution to current knowledge? A scientific discussion of the strengths and limitations associated with these case reports is needed.

The authors believe that the use of intrasocket granulation tissue for flap closure is not a widely accepted practice; however, the technique can be useful in limited cases where the IGT is thick. (lines 292-297) Also, within the limitations of this case report, authors do suggest a well-designed study to further investigate this technique. (lines 300-301)

- The patient should share their perspective on the treatment they received in one to two paragraphs—patient-reported outcomes (PROs) — what the patient reports about their treatment experience.

As the objective of this case report was to describe the technique itself, PROs have not been included in the project.   

Reviewer 3 Report

In my opinion, this surgical technique's rationale is incorrect because the granulation tissue is not mature and healthy and therefore is unsuitable for regeneration.

Therefore it is not correct to use this tissue for closure and regeneration also this tissue is too loose and it is not possible for good suturing. The format of the paper is ok and I didn't find any problem.

Author Response

Thank you very much for your comments.  Below is the response to address your concern.

In my opinion, this surgical technique's rationale is incorrect because the granulation tissue is not mature and healthy and therefore is unsuitable for regeneration.

Therefore it is not correct to use this tissue for closure and regeneration also this tissue is too loose and it is not possible for good suturing. The format of the paper is ok and I didn't find any problem.

The authors believe that the use of intrasocket granulation tissue for flap closure is not a widely accepted practice; however, the technique can be useful in limited cases where the IGT is thick and without inflammation. (lines 292-300). 

Round 2

Reviewer 2 Report

I agree to have the manuscript revised.

Author Response

Thank you very much for your approval.